# Impact of Iron on the Fe–Co–Ni Ternary Nanocomposites Structural and Magnetic Features Obtained via Chemical Precipitation Followed by Reduction Process for Various Magnetically Coupled Devices Applications

**DOI:** 10.3390/nano11020341

**Published:** 2021-01-29

**Authors:** Tien Hiep Nguyen, Gopalu Karunakaran, Yu.V. Konyukhov, Nguyen Van Minh, D.Yu. Karpenkov, I.N. Burmistrov

**Affiliations:** 1Department of Functional Nanosystems and High-Temperature Materials, National University of Science and Technology “MISiS”, 119049 Moscow, Russia; htnru7@yandex.ru (T.H.N.); glas100@yandex.ru (I.N.B.); 2Department of Materials Science and Engineering, Le Quy Don Technical University, Hanoi 100000, Vietnam; 3Biosensor Research Institute, Department of Fine Chemistry, Seoul National University of Science and Technology (Seoul Tech), Seoul 01811, Korea; 4Institute of Research and Development, Duy Tan University, Danang 550000, Vietnam; 5Institute of Technology, Hanoi 100000, Vietnam; 6Faculty of Physics, Lomonosov Moscow State University, GSP-1, Leninskie Gory, 119991 Moscow, Russia; karpenkov_d_y@mail.ru; 7Engineering Centre, Plekhanov Russian University of Economics, Moscow 117997, Russia

**Keywords:** Fe–Co–Ni nanocomposites, ternary nanocomposites, chemical precipitation, magnetic materials

## Abstract

This paper presents the synthesis of Fe–Co–Ni nanocomposites by chemical precipitation, followed by a reduction process. It was found that the influence of the chemical composition and reduction temperature greatly alters the phase formation, its structures, particle size distribution, and magnetic properties of Fe–Co–Ni nanocomposites. The initial hydroxides of Fe–Co–Ni combinations were prepared by the co-precipitation method from nitrate precursors and precipitated using alkali. The reduction process was carried out by hydrogen in the temperature range of 300–500 °C under isothermal conditions. The nanocomposites had metallic and intermetallic phases with different lattice parameter values due to the increase in Fe content. In this paper, we showed that the values of the magnetic parameters of nanocomposites can be controlled in the ranges of M_S_ = 7.6–192.5 Am^2^/kg, M_r_ = 0.4–39.7 Am^2^/kg, M_r_/M_s_ = 0.02–0.32, and H_cM_ = 4.72–60.68 kA/m by regulating the composition and reduction temperature of the Fe–Co–Ni composites. Due to the reduction process, drastic variations in the magnetic features result from the intermetallic and metallic face formation. The variation in magnetic characteristics is guided by the reduction degree, particle size growth, and crystallinity enhancement. Moreover, the reduction of the surface spins fraction of the nanocomposites under their growth induced an increase in the saturation magnetization. This is the first report where the influence of Fe content on the Fe–Co–Ni ternary system phase content and magnetic properties was evaluated. The Fe–Co–Ni ternary nanocomposites obtained by co-precipitation, followed by the hydrogen reduction led to the formation of better magnetic materials for various magnetically coupled device applications.

## 1. Introduction

Presently, research on magnetic material is one of the most promising fields. Continuous research on the improvement of magnetic materials is carried out regularly. The magnetic material based on coercive field magnitude can be divided into several groups: soft magnetic, hard magnetic, and materials with intermediate coercivity, classified as a particular class of semi-hard materials [1]. The major applications are magnetically coupled devices (brakes, clutches, tensioners), relay magnets, magnetic tooling holders, magnetic chucks, flip signs, sensor magnets, magnetic stirrer bars, and level sensors [2]. 

In turn, transition 3-d metals are widely used for the production of magnetic hybrid composites including those based on nano dispersed components [3,4,5,6]. Due to the high surface area of nano dispersed particles and their small size compared to the electron path length, their physical properties differ significantly from bulk materials [7,8]. As a result, nanomaterials with the size of structural elements close to the size of a single domain have excellent magnetic characteristics [9,10].

The most widely used materials are composed of Fe, Co, and Ni individually or in the form of composites [11,12,13,14,15]. However, up to today, the combination of the Fe–Co–Ni ternary system is not understood properly. Magnetic nanopowders have a very high potential for application as nano dispersed fillers of new functional composite materials such as polymer-bonded magnets (magnetoplastics), magnetic coatings, rubbers, paints, adhesives, etc. [16,17,18]. Magnetoplastics are highly commercially attractive for creating various parts, electronic elements, and devices with semi-hard magnetic properties due to their low processing costs, the ability to create flexible and stretchable pieces, parts with complex shapes, corrosion resistance, etc. Furthermore, the magnetic nanopowders are excellent electromagnetic wave absorbers due to their high surface area [19,20,21]. Radio-absorbing materials are used for the electromagnetic compatibility of electronic devices, protection of computer information processing systems from unauthorized access, protection of biological objects from electromagnetic radiation, etc. The nano dispersed magnetic filler plays a crucial role in such composite materials. Thus, developing the production methods of such materials, which provide controlled magnetic properties, is an essential scientific and technical task.

Following this, it is possible to control the nanoscale magnetic material characteristics by changing the size, shape, composition, and structure of nanocomposites within certain limits [3,4,5,6,11,12,13,14,15]. In turn, not all methods for nanopowder production allow for controlling their properties during the process. The chemical precipitation method of nanocomposite production fully meets the requirements. Despite its multi-step approach, this method allows the properties of the product to be controlled at each stage of its production. Therefore, the method allows chemically pure nanopowders to be produced with a given composition, shape, and dispersion [22,23].

This paper aims to study the peculiarities of the synthesis of Fe–Co–Ni ternary nanocomposites through the chemical precipitation method followed by reduction, and to understand the impact of reduction temperature and chemical composition on the phase formation, structure, particle size distribution, and magnetic features of the of synthesized Fe–Co–Ni ternary nanocomposites.

## 2. Materials and Methods

The ternary Fe–Co–Ni nanocomposites were prepared by the chemical precipitation method followed by reduction. The production scheme is presented in Figure 1.

The chemically pure nitrate salts Fe(NO_3_)_3_·9H_2_O (≥99.95%), Co(NO_3_)_2_·6H_2_O (≥98%), and Ni(NO_3_)_2_·6H_2_O (≥99%) (Russian standard, produced by “AO Vekton”) were used as precursors for the synthesis. Three mixtures of nitrate water solutions were prepared with the following Fe–Co–Ni mole ratio of (1:1:1); (3:1:1), and (5:1:1) to produce the desired nanocomposites. The special facility “Nanochem” (MISiS, Moscow, Russia) was used for the co-precipitation of hydroxides in an automatic regime at constant pH = 11 and continuous mixing. The facility consists of a reactor made of glass with a 5-liter volume, a pH-meter (Mettler Toledo, Ohio, USA), two peristaltic pumps, an electric mixer, and a PC controller. The precipitation was carried out by 10 mass. % alkali (NaOH) at room temperature. The chemical reaction was given below for the formation of precursor hydroxides.
Me(NO_3_)_x_ + xNaOH = Me(OH)_x_ + xNaNO_3_(1)
where Me is the cation Fe, Ni, Co, and x is the metal valence.

The high-speed centrifuge Rotixa 50S (Hettich, Kirchlengern, Germany) was applied to wash the sedimented precipitate. The hydroxides, after washing, were dried in a drying chamber at 40 °C for 72 h. The hydrogen reduction of dried samples was carried out in the tube furnace NABERTHERM RSR 120-750/11 (Nabertherm, Lilienthal, Germany) for 3 h at 300, 400, and 500 °C. The hydrogen supplier was a non-alkaline hydrogen generator SAM-1 (Etalonpribor, Moscow, Russia) with a capacity of 80 L/h with a purity of 99.9%. The hydrogen passed through an additional silica-based drying system before entering the quartz reactor of the furnace. The passivation of the reduced samples was carried out in a flow of technical nitrogen, which contained no more than 0.4 percent by volume of oxygen. 

After passivation, the nanocomposites were analyzed using different methods.

The chemical and phase composition of the nanocomposites were identified via x-ray powder diffraction (XRD) analysis utilizing a Difray-401 (Scientific Instruments JSC, Saint Petersburg, Russia) with a Cr-Kα radiation source in Bragg Brentano configuration, varying the 2*θ* angle from 20 to 140°. The XRD patterns of the obtained samples were refined using Match 3 and Base-Diffract programs with the International Centre for Diffraction Data (ICDD) standard date files Joint Committee on Powder Diffraction (JCPDS). The lattice parameters were calculated based on the experimental values of the interplanar spacing’s using the extrapolation function of Taylor–Sinclair [24]. The presence of elements was confirmed by energy-dispersive x-ray spectroscopy (EDX) using the attached EDX SSD detector (X-MAX, Hitachi, Tokyo, Japan).

Thermal gravimetric (TG) measurements were carried out using SDT-Q600 (TA Instruments, New Castle, USA) in nonisothermal conditions under hydrogen flow, with a heating rate of 10 °C/min in the range from 25 to 700 °C. Scanning electron microscope SEM TESCAN VEGA3 SB (TESCAN, Brno, Czech Republic) and transmission electron microscope TEM LEO 912 AB OMEGA (Carl Zeiss, Oberkochen, Germany) techniques were used to figure out the size and shape of nanocomposites. Selected area electron diffraction (SAED) patterns were obtained during transmission electron microscopy (TEM) investigations. The NOVA 1200 (Quantachrome Instruments, Boynton Beach, USA) was used for specific surface area (*Sa*) measurements. The calculation of the *Sa* was carried out by the Brunauer–Emmett–Teller (BET) model [25]. The average particle size was calculated using the equation [26]:(2)$$D=\frac{6}{\rho \cdot S_{a}}$$
where *ρ* is the density of the material in kg/m^3^. 

The magnetic properties of the nanocomposites were analyzed on a vibrating sample magnetometer (7410 Series VSM; Lake Shore Cryotronics, Westerville, OH, USA).

## 3. Results and Discussion

### 3.1. Phase Formation of Fe–Co–Ni Nanocomposites

The TG kinetic curves of the 1Fe–Co–Ni, 3Fe–Co–Ni, and 5Fe–Co–Ni hydroxides are presented in Figure 2 and Appendix A. The obtained TG data represent the presence of crystallohydrates (FeOOH·xH_2_O; Co(OH)_2_·yH_2_O; Ni(OH)_2_·zH_2_O) [27]. The coefficients x, y, z are variables and depend on the precipitation and drying conditions as well as the dispersity of the hydroxides. The dehydration processes of iron, nickel, and cobalt hydroxides occur in the temperature range of 25–130 °C (Figure 2). The formation of oxides occurs in the temperature range of 130–360 °C. However, the formation of intermetallic phases has occurred in the temperature range of 300–450 °C. The phase composition and variation of the final products are mainly formed due to the reduction process. Thus, in this research, we focused on the processes of the reduction stages. Figure 2a represents the 1Fe–Co–Ni sample TG curve, where two broad peaks were observed, which represents that the dehydration of hydroxides occurred between 25–210 °C. In addition, a weight loss was observed gradually. Between 210 and 300 °C, another peak, along with weight loss, was observed, which is due to the formation of the oxide phase of the composites. A further increase in temperature from 300–420 °C led to two peaks, which is due to the intermetallic phase formation. Figure 2b,c represents the TG curve of 3Fe–Co–Ni and 5Fe–Co–Ni samples. The major five peaks observed in both samples were almost similar. However, the variation in the intensity of the peaks was slightly different. The dehydration processes of hydroxides occur in the temperature range of 25–200 °C. Around 200–300 °C, a single peak along with weight loss was observed, which is due to the formation of the oxide phase of the composites. Additionally, a further increase in temperature from 300–420 °C led to two peaks, which was due to the intermetallic phase formation. The reduction process of the 1Fe–Co–Ni was fully completed at 430 °C. This was 100 degrees lower in comparison with 3Fe–Co–Ni and 5Fe–Co–Ni samples. This can be explained by the formation of intermediate Fe_2_O_3_·NiO, Fe_2_O_3_·CoO spinel phases during the metallization process. Metallization refers to the processes that occur during the dehydration and reduction of oxygen-containing compounds. Nickel and cobalt oxides have lower reduction temperatures than iron oxides [28,29]. Thus, the spinel break leads to the activation of iron oxide reduction. An increase in iron content leads to the stabilization of iron oxides, which are usually reduced in the range of 400–550 °C [23]. The peaks of 3Fe–Co–Ni and 5Fe–Co–Ni on the Thermogravimetry (TG) curves were stretched into the region of high temperatures.

1Fe–Co–Ni, 3Fe–Co–Ni, and 5Fe–Co–Ni nanocomposites were reduced under a hydrogen environment at various temperatures such as 300, 400, and 500 °C, respectively, and the obtained XRD patterns are shown in Figure 3. The 300 °C reduced samples contained oxide phases of γ–Fe_2_O_3_, Fe_2_O_3_·NiO, and Fe_2_O_3_·CoO [30]. This result is exactly matched with the TG results as we have observed that the oxide formation takes place due to the dehydration of nanocomposites. The samples after 400 °C reduction contained intermetallic and oxide faces, which are also well matched with the TG results. After 500 °C reduction, all the XRD patterns showed the formation of metallic 1Fe–Co–Ni nanocomposites.

The experimentally determined values of interplanar distances were used to calculate the lattice parameters of the Fe–Co (bcc) and Fe–Ni (fcc) phases using the Taylor–Sinclair extrapolation function [24]. The results are summarized in Table 1 and Appendix A. It is shown that when the nanocomposites reduced at 300 and 400 °C had an increase in iron content, the lattice period of the bcc- and fcc-nanocomposites increased. In different studies [31,32,33], the authors revealed the dependence of the value of the crystal lattice period of nanocomposites in the nanostructured systems Fe–Co–Ni. It is shown that in these systems, the lattice period of the bcc- and fcc-nanocomposites increased with an increase in the iron and cobalt content. At the same time, it decreased with an increase in nickel content. These observations were in good agreement with the result we obtained in this work. 

In contrast, the nanocomposites reduced at 300 and 400 °C showed less variation than that of samples obtained at 500 °C. The samples obtained at 500 °C showed variation in their lattice parameters. The maximum values of a = 0.2865 nm for the bcc phase and a = 0.3586 nm for the fcc phase were observed when the ratio of the number of iron atoms, cobalt, nickel was equal to 3:1:1. A further increase in the iron content up to the 5:1:1 composition led to a decrease in the lattice parameter down to 0.2860 nm for the bcc and 0.3580 nm for the fcc phase. 

The composition of the forming phases of the 1Fe–Co–Ni nanocomposites was estimated from the change in the lattice parameters of the Fe–Co and Fe–Ni solid mixture based on the α-Fe bcc and the Ni fcc lattice, respectively [34]. Table 1 shows that with an increase in the Fe addition, the Fe–Co and Fe–Ni solid mixture compositions also changed toward an increase in iron. The exception was the 3Fe–Co–Ni 500 °C sample, where both solid mixtures contained much more iron than the other samples.

The final scheme of formation of the phase composition is shown in Figure 4. It should be noted that the reduction was carried out at a temperature of 500 °C, so magnetite was directly reduced to iron.

The results of the EDX analysis are presented in Table 1 and Appendix A. All nanocomposites contained oxygen in different amounts. The data were converted to pure metals to simplify the comparison. As above-mentioned, the actual metal content was in good agreement with the calculated one. The amount of oxygen in the sample Fe–Co–Ni reduced at 300 °C was 18.2 mass. % according to the EDX analysis. Thus, the total quantity of the spinel phases was about 67 mass. %.

### 3.2. Dispersion and Morphological Characteristics

The specific surface area and the average particle size of the samples estimated by the BET method are shown in Table 2. The samples reduced at 300 °C had significantly higher *S_a_* due to the presence of intermediate oxide phases: γ–Fe_2_O_3_, Fe_2_O_3_·NiO, Fe_2_O_3_·CoO. An increase in reduction temperature caused a significant decrease in the *S_a_* values due to coagulation sintering processes. It was observed that the increase in iron in the nanocomposites led to an increase in the surface area and a reduction in the average particle size. The reason may be due to the formation of iron nanoparticles. Furthermore, it was also observed that the surface area of the sample was reduced, and the average particle size was increased with the increase in the reduction temperature. This was due to the metallic phase formation. 

The SEM, TEM, and SAED analysis of the Fe–Co–Ni nanocomposites are presented in Figure 5, Appendix A. The nanocomposites of the Fe–Co–Ni sample reduced at 300 °C were aggregated (Figure 5a). This means that the particles could be separated by a mechanical or physical method (ultrasound, for example). An increase in the reduction temperature from 300 °C to 400 °C caused an increase in the average particle size (D_SEM_) from 44 to 55 nm. The nanocomposites reduced at 400 °C were partly in the first stage of sintering (Figure 5c) with spherical particles. The micrograph showed narrow ranges between some of the particles. An increase in the reduction temperature to 500 °C resulted in a more significant rise of D_SEM_ to 66 nm (Figure 5e), with higher aggregated bigger spherical particles. The possible mechanism behind the changes in the particle size would be from the XRD results, surface area, and average particle size. The XRD results (Figure 3) showed that with the increase in temperature, the particles transformed from a semi amorphous structure to more crystalline structures with the reduction process, which resulted in the intermetallic phases. From Table 2, the surface area results easily show that the surface was reduced from 20.3 m^2^/g to 1.9 m^2^/g with the increase in temperature, which was also the reason for the formation of intermetallic phases and the interaction of smaller particles to form bigger particles. Hence, the temperature influence was the increase in the average size of the particles. All particles had wide ranges, and there were noticeable signs of the second stage of sintering when the nuclei began to converge. It should be noted that the shape of the particles changed from a spherical to faceted shape with increasing reduction temperature. The TEM photographs and SAED patterns of the reduced Fe–Co–Ni compound (Figure 5b,d,f) also correlated with the results of the SEM, BET, and phase analysis.

### 3.3. Magnetic Properties

Hysteresis loops of investigated nanopowders are presented in Figure 6. The nanocomposite samples reduced at 500 °C possessed low coercivity, hence it revealed that the obtained samples were soft magnetic materials. In contrast, the nanopowders, synthesized at lower temperatures, were semi-hard magnetic materials.

The detailed magnetic features of the nanocomposites in terms of remanence ratio (*M*_r_/*M*_s_), saturation magnetization *M*_s_, coercive field *H*_cM_, and remanent magnetization *M*_r_, are represented in Table 3. The formation of metallic and intermetallic phases under reduction resulted in the increase in the saturation magnetization value of nanopowders due to higher magnetic moments of the former compared to ferritic phases [35]. With the increasing reduction temperature, all nanocomposite saturation magnetization increased due to their reduction degree. The EDX data prove these results. The residual oxygen content in the samples significantly decreased under the increasing reduction temperature, due to which the ferritic phases were converted into the intermetallic and metal phases. 

Moreover, increasing the reduction temperature accelerated the particle size growth and caused a more perfect crystal structure of magnetic nanopowders, which additionally increased the magnetization value. The increased particle size resulted in the reduction in surface atom number, which contributes to the misaligned constituent of spin magnetic moment and gains the magnetization of the material as a whole [11,12,13,36,37].

For the samples of all series, the increasing Fe content initially led to an increase in saturation magnetization value, and after reaching the maximum, the fall in the parameter occurred. The 3Fe–Co–Ni sample had the highest saturation magnetization within the investigated series. The reduction process of magnetic nanopowders at low temperatures led to maximum remanence magnetization and coercivity values. However, with further increase in the reduction temperature, a decrease in these parameters was observed.

It is known that the coercivity *H_c_* of ferromagnetic nanocomposites is highly dependent on their size. In particular, it can be defined with the following equation $H_{c}=a+b/d$ with *d* > *d*_s_ and $H_{c}=g-h/d$ with *d* > *d*_s_ [38,39,40], wherein *d*_s_ is the single-domain critical diameter and *a*, *b*, *g*, and *h* are constants. Furthermore, it results in *H*_c_ peaks around the point *d*_s_. Additionally, the values of the single-domain critical diameter *d*_s_ are specified as follows [41]:(3)$$d_{s}=\frac{9\cdot \sigma_{w}}{2\pi \cdot M_{s}^{2}}$$
where *σ**_w_* is the domain wall energy density, which can be defined by the expression:(4)$$\sigma_{w}={\left(\frac{2k_{B}\cdot T_{C}\cdot \left|K_{1}\right|}{a}\right)}^{\frac{1}{2}}$$
where $T_{C}$ is the Curie temperature; $\left|K_{1}\right|$ is the magnetocrystalline anisotropy constant; *a* is the lattice constant; and *k*_B_ is the Boltzmann constant.

According to the above model, the *d*_s_ value of Fe–Co–Ni nanocomposites ranges between 5 and 60 nm. As the particle size was reduced at a temperature of about 300 °C, nanocomposites had a single-domain structure. The samples reduced at 400 and 500 °C possessed average particle sizes bigger than *d_s_*. Consequently, the increasing reduction temperature caused an increase in nanoparticle size, which degraded the coercivity of the nanopowders. Moreover, the reduction in the crystal structure defects, deformation stress, and degree of disorder under increasing average particle size affected the magnetic parameter values [41,42,43,44]. In turn, the magnitude of the coercive field can be defined using Brown’s equation [45,46]:(5)$$H_{c}=H_{a}=\frac{2K}{\mu_{B}\cdot M_{s}}$$
where *K* is the magnetocrystalline anisotropy constant and $H_{a}$ is the anisotropy field. Thus, coercivity is inversely proportional to saturation magnetization.

## 4. Conclusions

Through the chemical precipitation method followed by the reduction process, we obtained three Fe–Co–Ni ternary nanocomposites. It was found that during the hydrogen reduction process of the Fe–Co–Ni ternary systems, intermediate spinel phases of Fe_2_O_3_·NiO and Fe_2_O_3_·CoO were formed. It was revealed that for Fe–Co–Ni nanocomposites reduced at 300 and 400 °C, with an increase in the iron content, the lattice constant (a) of the bcc and fcc phase increased, and the content of cobalt in the bcc phase and nickel in the fcc phase decreased. It is noted that the reduction temperature did not significantly affect the crystal lattice parameters of the nanocomposites. In the case of ternary compositions obtained at 500 °C, the dependence of the lattice parameter of the solid mixture on the iron content had an extreme character. The maximum values of a = 0.2865 nm for the bcc phase and a = 0.3586 nm for the fcc phase were observed when the ratio of the number of iron, cobalt, nickel atoms was equal to 3:1:1. A further increase in the iron content up to the 5:1:1 composition led to a decrease in the lattice parameter down to 0.2860 nm for bcc and 0.3580 nm for the fcc phase. It was shown that during the reduction of Fe–Co–Ni ternary compositions with hydrogen, values of the average particle size (D_SEM_) increased with an increase in the reduction temperature. The increase in reduction temperature from 300 °C to 400 and 500 °C caused increases in the average particle size from 44 to 55 and 66 nm, respectively. It was found that the average particle size of ternary compositions decreased with an increase in the iron content. Furthermore, we observed that the values of the magnetic parameters of nanopowders could be controlled in the large ranges of M_S_ = 7.6–192.5 Am^2^/kg, M_r_ = 0.4–39.7 Am^2^/kg, M_r_/M_s_ = 0.02–0.32, and H_cM_ = 4.72–60.68 kA/m via regulating of the composition and reduction temperature of the formers. Within the reduction process, a drastic increment in the magnetic features is observed because of intermetallic and metallic phase formation. The variation in magnetic characteristics is caused by the degree of reduction, particle size growth, and crystallinity enhancement. Moreover, diminishment of the surface spins fraction of the nanoparticles under their growth induced an increase in the saturation magnetization. The obtained Fe–Co–Ni ternary nanocomposites at 300 °C reductions were observed to possesses the highest value of coercivity *H*_c_. Thus, this nanocomposite material can be an excellent substitute for various commercially available nanocomposites for better device applications.

## Figures and Tables

**Figure 1 nanomaterials-11-00341-f001:**
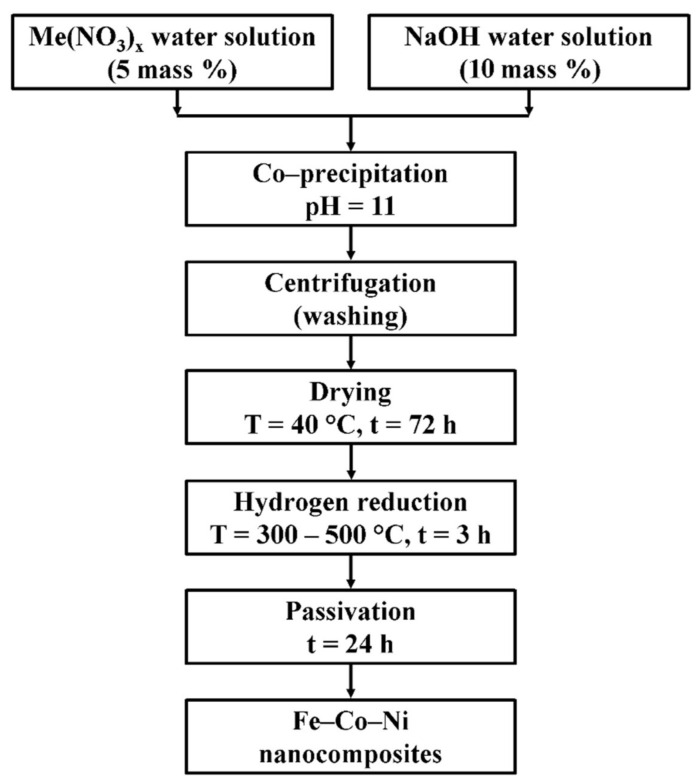
The production scheme of Fe–Co–Ni nanocomposites.

**Figure 2 nanomaterials-11-00341-f002:**
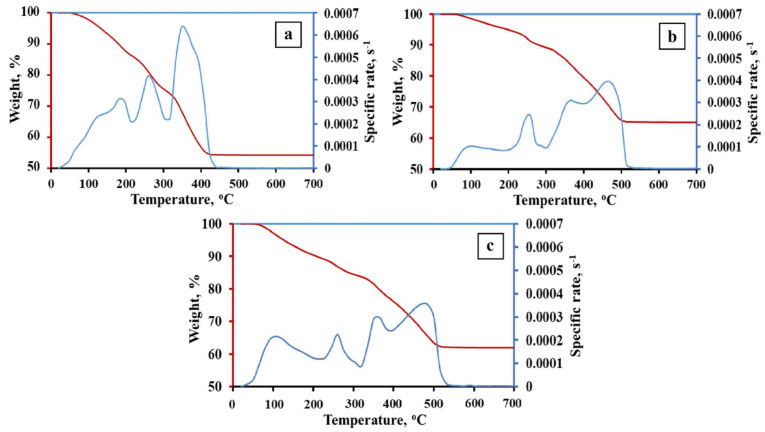
Thermal gravimetric (TG) kinetic curves of the Fe–Co–Ni hydroxides. (**a**) 1Fe–Co–Ni, (**b**) 3Fe–Co–Ni, (**c**) 5Fe–Co–Ni, (*dm/dτ*)/*m*_0_ is the specific rate; *m*_0_ is the initial weight.

**Figure 3 nanomaterials-11-00341-f003:**
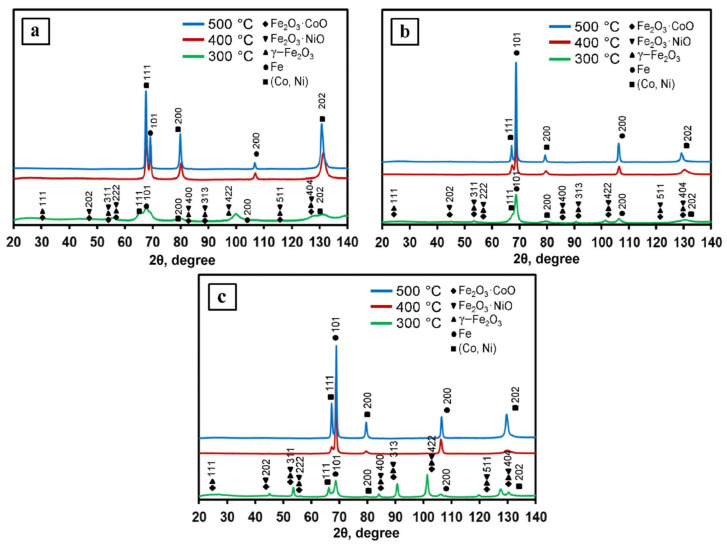
The Fe–Co–Ni nanocomposites’ x-ray diffraction (XRD) patterns reduced at 300–500 °C. (**a**) 1Fe–Co–Ni, (**b**) 3Fe–Co–Ni, (**c**) 5Fe–Co–Ni.

**Figure 4 nanomaterials-11-00341-f004:**
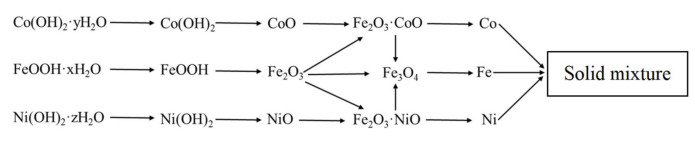
Scheme of phase formation in Fe–Co–Ni magnetic nanocomposites between 300–500 °C.

**Figure 5 nanomaterials-11-00341-f005:**
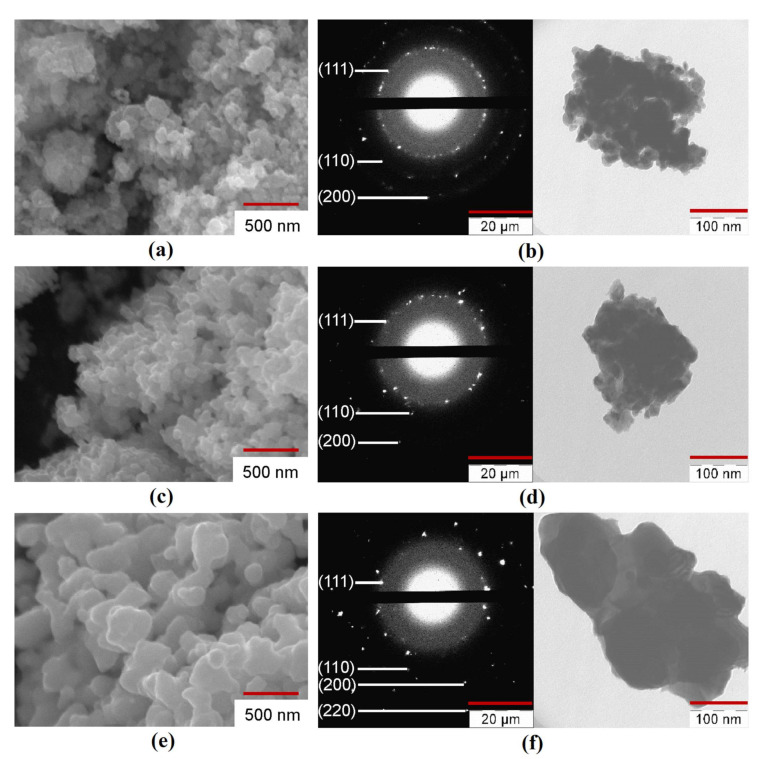
Scanning electron microscope (SEM), transmission electron microscope (TEM) images, and Selected area electron diffraction (SAED) patterns of Fe–Co–Ni nanocomposites (**a**,**b**) 1Fe–Co–Ni T_R_ = 300 °C, (**c**,**d**) 3Fe–Co–Ni T_R_ = 400 °C, (**e**,**f**) 5Fe–Co–Ni T_R_ = 500 °C.

**Figure 6 nanomaterials-11-00341-f006:**
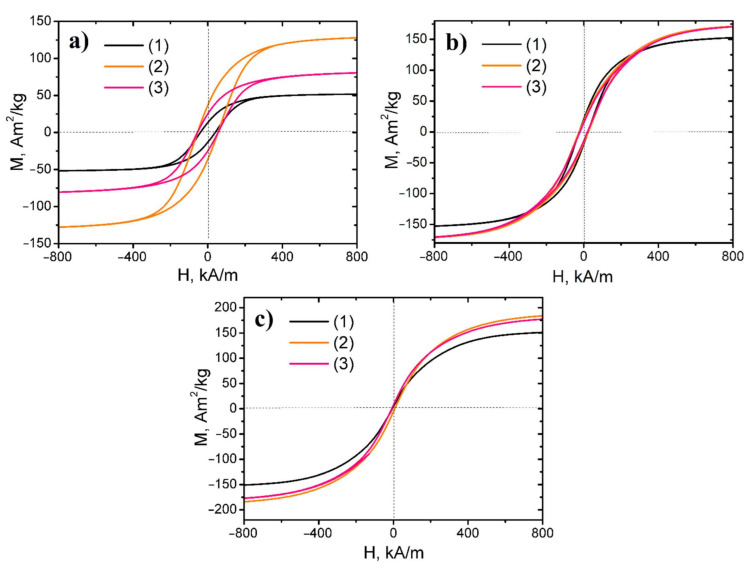
Hysteresis loops of Fe, Ni, and Co ternary system-based magnetic nanopowders: (**a**) reduced at 300 °C series, (**b**) reduced at 400 °C series, (**c**) reduced at 500 °C series (1) Fe–Co–Ni samples, (2) 3Fe–Co–Ni samples, (3) 5Fe–Co–Ni samples.

**Table 1 nanomaterials-11-00341-t001:** Co and Ni content in iron-based nanocomposites.

№	Sample	Reduction Temperature, °C	Element Composition by EDX, mass.%.			Lattice Period *a*, nm		Content of Co in bcc Phase, at. %	Content of Ni in fcc Phase, at.%.
			Fe	Co	Ni	bcc	fcc		
1	1Fe–Co–Ni	300	32.3	34.9	32.8	–	0.3555	–	74.5
2	3Fe–Co–Ni					0.2864	0.3568	30	62
3	5Fe–Co–Ni					0.2870	–	~20	–
4	1Fe–Co–Ni	400	57.9	21.2	20.9	0.2854	0.3554	52	74.5
5	3Fe–Co–Ni					0.2864	0.3569	30	62
6	5Fe–Co–Ni					0.2868	0.3577	15	60
7	1Fe–Co–Ni	500	70.5	15.1	14.4	0.2856	0.3563	48	63
8	3Fe–Co–Ni					0.2865	0.3586	25	31
9	5Fe–Co–Ni					0.2860	0.3580	40	59

**Table 2 nanomaterials-11-00341-t002:** Results of measuring the specific surface area and the average particle size by the Brunauer–Emmett–Teller (BET) method.

No.	Sample	Reduction Temperature, °C	Specific Surface Area *S_a_*, m^2^/g	Average Size of Particles D, nm
1	1Fe–Co–Ni	300	20.3	54
2	3Fe–Co–Ni		24.2	49
3	5Fe–Co–Ni		53.3	36
4	1Fe–Co–Ni	400	5.9	77
5	3Fe–Co–Ni		7.0	75
6	5Fe–Co–Ni		8.5	72
7	1Fe–Co–Ni	500	0.5	151
8	3Fe–Co–Ni		1.2	135
9	5Fe–Co–Ni		1.9	115

**Table 3 nanomaterials-11-00341-t003:** Magnetic parameters of investigated Fe–Co–Ni nanocomposites.

Sample	Reduction Temperature, °C	Magnetic State, Ferro/para	*M*_s_, A m^2^ kg^−1^ (emu/g)	*M*_r_, A m^2^ kg^−1^ (emu/g)	*M*_r_/*M*_s_	*H*_cM_, kA/m
1Fe–Co–Ni	300	Ferromagnetic	52.3	14.8	0.28	41.6
3Fe–Co–Ni		Ferromagnetic	130.9	39.7	0.30	60.16
5Fe–Co–Ni		Ferromagnetic	84.1	27.0	0.32	60.64
1Fe–Co–Ni	400	Ferromagnetic	155.8	25.1	0.16	24.32
3Fe–Co–Ni		Ferromagnetic	176.6	21.5	0.12	26.32
5Fe–Co–Ni		Ferromagnetic	176.6	17.6	0.1	21.84
1Fe–Co–Ni	500	Ferromagnetic	156.3	4.8	0.03	4.72
3Fe–Co–Ni		Ferromagnetic	192.5	6.2	0.03	5.04
5Fe–Co–Ni		Ferromagnetic	185.2	3.7	0.02	5.92

## Data Availability

The data presented in this study are available on request from the corresponding author.

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
