# Peer review of "Impact of Iron on the Fe–Co–Ni Ternary Nanocomposites Structural and Magnetic Features Obtained via Chemical Precipitation Followed by Reduction Process for Various Magnetically Coupled Devices Applications"

_nanomaterials, 2021, doi:10.3390/nano11020341_

Round 1

Reviewer 1 Report

  • The English text has to be edited thoroughly by a professional company and the confirmation of this company has to be submitted prior to publication
  • Use larger and clear lettering in the figures
  • What do the 20µm mean in the diffraction patterns of Fig. 5?
  • The numerous repetitions should be removed
  • A more concise formulation is desirable

Author Response

Response to Reviewer 1 Comments

Point 1: The English text has to be edited thoroughly by a professional company and the confirmation of this company has to be submitted prior to publication

Response 1: We thank the reviewer for his/her valuable comments on our manuscript. We have addressed all the critiques raised by reviewer and hope that our effort will satisfy the reviewer. The manuscript is revised the English text with the help of the native speaker and highlighted the changes in the revised manuscript.

Point 2: Use larger and clear lettering in the figures

Response 2: As per the suggestion, figures are revised with larger and clear letters in the revised manuscript.

Point 3: What do the 20µm mean in the diffraction patterns of Fig. 5?

Response 3: The 20µm represents the scale of measurement in diffraction pattern in Fig. 5.

Point 4: The numerous repetitions should be removed

Response 4: The repetitions are avoided.

Point 5: A more concise formulation is desirable

Response 5: The concise formulation is presented.

Reviewer 2 Report

A comprehensive characterization of nanopowders composed from three fundamental elements responsible for magnetism is provided in the paper. The paper discusses in detail the issues related to production of nanocomposites and the effect of the choice of reduction temperature on magnetic properties of the developed compounds. The paper is well written  and it deserves publication in the present form. The authors are advised to improve the quality of some photos, e.g. Figure 6 is hardly legible. With regard to the aforementioned Figure, the authors might consider showing the hysteresis curves for a more limited range of magnetic field H, e.g. <-800 kA/m; 800 kA/m> (half of the one originally presented) since in saturation region the hysteresis curves practically dissapear and thus a lot of space is wasted in the Figure in the present form. 

The authors might include a citation of the original paper by Brunauer, Emmett and Teller (J. Am. Chem. Soc. 1938, 60, 2, 309–319) when they mention the BET model, since the reviewer believes not all readers are familiar with the concept. 

Could the authors provide a reference to the formula (1) which they used for computing the average particle size e.g. F. Michel & L. Courard (2014) Particle Size Distribution of Limestone Fillers: Granulometry and
Specific Surface Area Investigations, Particulate Science and Technology: An International Journal, 32:4, 334-340?

Author Response

Response to Reviewer 2 Comments

Point 1: A comprehensive characterization of nanopowders composed from three fundamental elements responsible for magnetism is provided in the paper. The paper discusses in detail the issues related to production of nanocomposites and the effect of the choice of reduction temperature on magnetic properties of the developed compounds. The paper is well written and it deserves publication in the present form.

 The authors are advised to improve the quality of some photos, e.g. Figure 6 is hardly legible. With regard to the aforementioned Figure, the authors might consider showing the hysteresis curves for a more limited range of magnetic field H, e.g. <-800 kA/m; 800 kA/m> (half of the one originally presented) since in saturation region the hysteresis curves practically disappear and thus a lot of space is wasted in the Figure in the present form. 

 Response 1: We thank the reviewer for the appreciation and positive comments on our manuscript. We have addressed all the critiques raised by reviewer and hope that our effort will satisfy the reviewer. As per the reviewer suggestion, the quality of the figure is improved. Also as per the suggestion the hysteresis curves are limited between the range of <-800 kA/m; 800 kA/m>.

Figure 6. Hysteresis loops of Fe, Ni, Co ternary system-based мagnetic nanopowders a – reduced at 300 °C series; b – reduced at 400 °C series; c – reduced at 500 °C series (1) – Fe-Co-Ni samples; (2) – 3Fe-Co-Ni samples; (3) – 5Fe-Co-Ni samples

Point 2: The authors might include a citation of the original paper by Brunauer, Emmett and Teller (J. Am. Chem. Soc. 1938, 60, 2, 309–319) when they mention the BET model, since the reviewer believes not all readers are familiar with the concept. 

Response 2: As per the reviewer suggestion the citation is included in the revised manuscript.

Point 3: Could the authors provide a reference to the formula (1) which they used for computing the average particle size e.g. F. Michel & L. Courard (2014) Particle Size Distribution of Limestone Fillers: Granulometry and
Specific Surface Area Investigations, Particulate Science and Technology: An International Journal, 32:4, 334-340?

Response 3: As per the reviewer suggestion the citation for average particle size is included in the revised manuscript.

Reviewer 3 Report

The work ‘Impact of iron on the Fe-Co-Ni ternary nanocomposites structural and magnetic features obtained via chemical precipitation followed by reduction process for various magnetically coupled devices applications’ presents the synthesis and study of the morphology, structural and magnetic properties of three different ternary metal nanomaterial samples, in which the content of iron was varied relative to the constant content of the other two metal elements. The manuscript is generally well written and the results are interesting, fulfilling the scope and subjects of the Nanomaterials Journal. The introduction and materials synthesis is well explained the references are adequate and the manuscript is generally easy to read.

However, there are some points in the manuscript that the authors should take into account and clarify.

  1. Lines 107-108: provide the level of purity for the reactant salts.
  2. Line 124: provide the purity of the hydrogen gas.
  3. Line 195: since Fe2O3 is (presumably) a spinel phase it should be named as γ-Fe2O3. The same amendment has to be done in the diagrams of Figure 3.
  4. Table 1, lists of Co in bcc and Ni in fcc contents: which method did the authors used to determine the exact Co and Ni content in the bcc and fcc phases? There is a general remark on using EDX analysis, but there is no specific explanation on that issue. This has to be described in the text. Moreover, there is no information on the relative mass content of each phase in the samples, which is important to explain further the magnetic properties as well. This could be obtained e.g. by Rietveld refinement of the XRD diagrams the authors already present.
  5. Table 1: typical error level values in all quantities should be specified, at least in the experimental section or given in the Table’s caption. For the lattice constant this is rather most important to determine if there is a difference in this parameter relative to the different content and reduction temperature, which is discussed further in the text. The same holds for Tables 2 and 3.
  6. Lines 254-255: the sole use of the expression ‘It is due to the metallic phase formation’ to explain the nature and morphology of nanoparticles in the samples is rather incomplete. These are significant characteristics that influence all other properties of the nanomaterials. The authors should give a more extensive explanation and description on the nature and morphology of the nanoparticles reduced at different temperatures with reference to the SEM images.

Lines 340-341: it is not clear what the authors what to express here, mainly by the phrase ‘… has a 100 C lower temperature …’. Please rephrase and clarify.

Line 344 (and wherever at the manuscript appears): in my opinion the expression 'lattice period' to describe the lattice constant or parameter is rather unfamiliar and perhaps misleading!

A few spelling or expression amendments:

Line 174: ‘occurs’

Lines 215-216: … show variation in their lattice …

Line 252: ‘… due to the iron nanoparticles formation.’

Line 253: ‘… sample is reduced …’

Line 254: ‘… reduction temperature.’

Line 296: ‘… EDX data prove these results.’

Line 301: ‘… increased the magnetization …’

Line 315 321, 322, 333: the equations have rather quite larger font sizes than the text.

Line 330: ‘… affect the magnetic parameters …’

Line 367: Rephrase ‘this material can be excellent materials nanocomposites’

Taking into account these suggestions in my opinion the work is suitable to be published in Nanomaterials Journal.

Author Response

Response to Reviewer 3 Comments

Point 1: The work ‘Impact of iron on the Fe-Co-Ni ternary nanocomposites structural and magnetic features obtained via chemical precipitation followed by reduction process for various magnetically coupled devices applications’ presents the synthesis and study of the morphology, structural and magnetic properties of three different ternary metal nanomaterial samples, in which the content of iron was varied relative to the constant content of the other two metal elements. The manuscript is generally well written and the results are interesting, fulfilling the scope and subjects of the Nanomaterials Journal. The introduction and materials synthesis is well explained the references are adequate and the manuscript is generally easy to read.

However, there are some points in the manuscript that the authors should take into account and clarify.

  1. Lines 107-108: provide the level of purity for the reactant salts.

Response 1: We thank the reviewer for his/her valuable comments on our manuscript. We have addressed all the critiques raised by reviewer and hope that our effort will satisfy the reviewer. As per the reviewer suggestion the purity for the reactant salts are included.

Point 2: Line 124: provide the purity of the hydrogen gas.

Response 2: The purity of hydrogen gas is included in the revised manuscript.

Point 3: Line 195: since Fe2O3 is (presumably) a spinel phase it should be named as γ-Fe2O3. The same amendment has to be done in the diagrams of Figure 3.

Response 3: As per the reviewer suggestion the changes are incorporated in the revised manuscript.

Point 4: Table 1, lists of Co in bcc and Ni in fcc contents: which method did the authors used to determine the exact Co and Ni content in the bcc and fcc phases? There is a general remark on using EDX analysis, but there is no specific explanation on that issue. This has to be described in the text. Moreover, there is no information on the relative mass content of each phase in the samples, which is important to explain further the magnetic properties as well. This could be obtained e.g. by Rietveld refinement of the XRD diagrams the authors already present.

Response 4: Thank you for the valuable question, the XRD analysis was used to identify the phases. The relative mass percent of each faces was difficult to identify hence, it is not incorporated in the manuscript. However, individual mass percentage is shown in Table 1 obtained by EDX analysis.

Point 5: Table 1: typical error level values in all quantities should be specified, at least in the experimental section or given in the Table’s caption. For the lattice constant this is rather most important to determine if there is a difference in this parameter relative to the different content and reduction temperature, which is discussed further in the text. The same holds for Tables 2 and 3

Response 5: As per the reviewer suggestion the error levels are displayed in Table 2 and 3.

Point 6: Lines 254-255: the sole use of the expression ‘It is due to the metallic phase formation’ to explain the nature and morphology of nanoparticles in the samples is rather incomplete. These are significant characteristics that influence all other properties of the nanomaterials. The authors should give a more extensive explanation and description on the nature and morphology of the nanoparticles reduced at different temperatures with reference to the SEM images.

Response 6: As per the reviewer’s suggestion the nature and morphology of nanoparticles explanation is provided in the revised manuscript.

Point 7: Lines 340-341: it is not clear what the authors what to express here, mainly by the phrase ‘… has a 100 C lower temperature …’. Please rephrase and clarify.

Response 7: As per the suggestion, the unclear sentences are removed from the Lines 340-341.

Point 8: Line 344 (and wherever at the manuscript appears): in my opinion the expression 'lattice period' to describe the lattice constant or parameter is rather unfamiliar and perhaps misleading!

Response 8: We thank the reviewer’s suggestion and the “lattice period” is changed to “lattice constant”.

Point 9: A few spelling or expression amendments:

Line 174: ‘occurs’

Lines 215-216: … show variation in their lattice …

Line 252: ‘… due to the iron nanoparticles formation.’

Line 253: ‘… sample is reduced …’

Line 254: ‘… reduction temperature.’

Line 296: ‘… EDX data prove these results.’

Line 301: ‘… increased the magnetization …’

Line 315 321, 322, 333: the equations have rather quite larger font sizes than the text.

Line 330: ‘… affect the magnetic parameters …’

Line 367: Rephrase ‘this material can be excellent materials nanocomposites’

Response 9: We thank the reviewer for his valuable suggestion and corrections. These correction and suggestion improved the quality of our paper, thank you. All the spelling errors and expression amendments are included in the revised manuscript.

Reviewer 4 Report

The manuscript needs revisions as follows.

The manuscript describes synthesis of Fe-Co-Ni nanocomposites by chemical precipitation and a reduction process. Chemical composition, reduction temperature greatly affects the phase formation, its structures, particle size distribution, and magnetic properties of Fe-Ni-Co nanocomposites.

The authors conclude as follows: Through the chemical precipitation method followed by the reduction process, we obtained three Fe-Co-Ni ternary nanocomposites. It was found that during the hydrogen reduction process of the Fe-Co-Ni ternary systems, intermediate spinel phases of NiO·Fe2O3 and CoO·Fe2O3 were formed. As a result, the material of the Fe-Co-Ni compositions has a 100°C lower temperature at the end of the reduction process than the compositions with higher iron contents (3Fe-Co-Ni and 5Fe-Co-Ni). It was revealed that for Fe-Co-Ni nanocomposites reduced at 300 and 400°C, with an increase in the iron content the lattice period (a) of the bcc, fcc phase increases, and the content of cobalt in the bcc phase, nickel in the fcc phase decreases. It is noted that the reduction temperature does not significantly affect the crystal lattice parameters of nanocomposites. In the case of ternary 346 compositions obtained at 500°C, the dependence of the lattice parameter of the solid mixture on the iron content has an extreme character. The maximum values of a = 0.2865 nm for the bcc phase, a = 0.3586 nm for the fcc phase are observed when the ratio of the number of iron, cobalt, nickel atoms equal to 3:1:1. A further increase in the iron content up to the 5:1:1 composition leads to a decrease in the lattice parameter down to 0.2860 nm for bcc and 0.3580 nm for the fcc phase. It is shown that during the reduction of Fe-Co-Ni ternary compositions with hydrogen, values of the average particle size (DSEM) increase with an increase in the reduction temperature. The increase of reduction temperature from 300 oC to 400 and 500 oC causes increases in the average particle size from 44 to 55 and 66 nm, respectively. It was found that the average particle size of ternary compositions decreases with an increase in the iron content. Further, we have observed that the values of magnetic parameters of nanopowders can be controlled in the large ranges of MS=7.6 –192.5 Am2/kg, Mr=0.4 – 39.7 Am2/kg, Mr/Ms=0.02 – 0.32, and HcM=4.72 – 60.68 kA/m via regulating of the composition and reduction temperature of the formers. Within the reduction process, a drastic increment in the magnetic features is observed because of intermetallic and metallic phase formation. The magnetic characteristics variation is caused by the degree of reduction, particle size growth, and crystallinity enhancement. Moreover, diminishment of surface spins fraction of the nanoparticles under their growth induced an increase of the saturation magnetization. The obtained Fe-Co-Ni ternary nanocomposites at 300 °C reductions were observed to possesses the highest value of coercivity Hc. Thus, this material can be excellent materials nanocomposites for various device applications

The authors mentioned as follows: The maximum values of a = 0.2865 nm for the bcc phase, a = 0.3586 nm for the fcc phase are observed when the ratio of the number of iron, cobalt, nickel atoms equal to 3:1:1. A further increase in the iron content up to the 5:1:1 composition leads to a decrease in the lattice parameter down to 0.2860 nm for bcc and 0.3580 nm for the fcc phase. It is shown that during the reduction of Fe-Co-Ni ternary compositions with hydrogen, values of the average particle size (DSEM) increase with an increase in the reduction temperature. The increase of reduction temperature from 300 oC to 400 and 500 oC causes increases in the average particle size from 44 to 55 and 66 nm, respectively. It was found that the average particle size of ternary compositions decreases with an increase in the iron content.

It is recommended to explain mechanism of change in particle size more clearly using a detailed crystal structure. You can use figures and calculations for easy understanding of readers.

The authors mentioned as follows: Further, we have observed that the values of magnetic parameters of nanopowders can be controlled in the large ranges of MS=7.6 –192.5 Am2/kg, Mr=0.4 – 39.7 Am2/kg, Mr/Ms=0.02 – 0.32, and HcM=4.72 – 60.68 kA/m via regulating of the composition and reduction temperature of the formers. Within the reduction process, a drastic increment in the magnetic features is observed because of intermetallic and metallic phase formation. The magnetic characteristics variation is caused by the degree of reduction, particle size growth, and crystallinity enhancement. Moreover, diminishment of surface spins fraction of the nanoparticles under their growth induced an increase of the saturation magnetization. The obtained Fe-Co-Ni ternary nanocomposites at 300 °C reductions were observed to possesses the highest value of coercivity Hc.

I recommend to explain the mechanism by which magnetic parameters are obtained in these materials. It is advisable to discuss the magnetic parameters in relation to the crystallographic characteristics, compositional characteristics, etc. of nano powders.

Author Response

Response to Reviewer 4 Comments

Point 1: It is recommended to explain mechanism of change in particle size more clearly using a detailed crystal structure. You can use figures and calculations for easy understanding of readers.

Response 1: We thank the reviewer for his/her valuable comments on our manuscript. We have addressed all the critiques raised by reviewer and hope that our effort will satisfy the reviewer. As per the reviewer suggestion the mechanism of change in particle size was discussed in details in the revised manuscript.

Point 2: I recommend to explain the mechanism by which magnetic parameters are obtained in these materials. It is advisable to discuss the magnetic parameters in relation to the crystallographic characteristics, compositional characteristics, etc. of nano powders.

Response 2: The magnetic properties of the nanocomposites were analyzed on the Vibrating Sample Magnetometer (7410 Series VSM; Lake Shore Cryotronics, USA). The magnetic results are presented as observed through magnetic analysis. The difference in the magnetic features are observed based on the difference in the particle size and phase of the nanocomposites.